# Current status of Tele-speech language therapy by type and support for patients with post-stroke aphasia: A scoping review

Yuhei Kodani[1,2], Shinsuke Nagami[3]*, Ayaka Yokozeki[4], Shinya Fukunaga[1], Katsuya Nakamura[1], Hikaru Nakamura[2]

**1** Department of Speech and Hearing Therapy, Faculty of Rehabilitation Sciences, Kawasaki University of Health and Welfare, Kurashiki, Okayama, Japan, **2** Graduate School of Health and Welfare, Okayama Prefectural University, Soja, Okayama, Japan, **3** Department of Communication Disorders, School of Rehabilitation Sciences, Health Sciences University of Hokkaido, Ishikari-gun, Hokkaido, Japan, **4** Department of Neurosurgery, Takamatsu Red Cross Hospital, Takamatsu, Kagawa, Japan

* shinsuke.nagami.0514@gmail.com

## Abstract

### Purpose

The purpose of this study was to classify and analyze trends in the assessment and training methods used in telepractice speech-language therapy (Tele-SLT) for people with aphasia (PWA), according to the type of Tele-SLT (synchronous, asynchronous, or combined). This study particularly aimed to identify gaps that prevent the establishment of Tele-SLT, a field that has gained significant attention post-COVID-19 pandemic.

### Design

A scoping review was conducted following the PRISMA-ScR guidelines.

### Setting and participants

Included were research articles on Tele-SLT for individuals aged 18 years or older diagnosed with post-stroke aphasia. Articles in both English and Japanese were reviewed, using five online databases (Medline, Embase, PsycInfo, Cochrane Library, and ICHUSHI Web).

### Methods

Studies involving Tele-SLT were categorized by support methods, content, study design, and outcomes. The quality of the extracted studies was also assessed. We also assessed the quality of the selected studies and performed a meta-analysis of some of the results.

### Results

Of the initial 1,484 articles, 35 met the eligibility criteria. Regarding Tele-SLT support methods, 3 articles (8.57%) focused on assessment methods, while 32 (91.43%) focused on training methods. Fourteen articles (40.00%) employed synchronous Tele-SLT delivery,

**Data availability statement:** All relevant data are within the paper and its Supporting information files.

**Funding:** This study, including research, writing, and publication processes, was supported by the following financial grants. Specifically, YK received funding from the Japan Society for the Promotion of Science (JSPS) KAKENHI (Grant Numbers 21K21154 and 24K13406), and HN received funding from the JSPS KAKENHI (Grant Number 23K09604). The funders had no role in study design, data collection and analysis, decision to publish, or preparation of the manuscript.

**Competing interests:** The authors declare that there is no conflict of interest.

20 (57.14%) employed asynchronous delivery, and 1 (2.86%) employed a combined approach. The methodological quality of 27 (77.14%) of the included Tele-SLT articles was rated as 'Low'. A meta-analysis of randomized controlled trials on Tele-SLT demonstrated that asynchronous training was effective for language function

## Conclusions and significance

This study highlights the need for more research, particularly on remote assessment and synchronous training methods, in Tele-SLT for PWA. Furthermore, this study emphasizes the need for improved research methodologies in this area. To provide high-quality support for PWA who have faced challenges accessing in-person speech-language therapy since the COVID-19 pandemic, further research and development of Tele-SLT implementation guidelines are needed

## Introduction

The incidence of stroke is increasing, especially among the elderly in an aging society [1,2]. As the number of stroke survivors increases, so too does the number of patients with aphasia (PWA), who face difficulties with language skills (listening, speaking, reading, and writing), communication (situational language use), well-being, and quality of life (QOL) [3–7]. Additionally, the COVID-19 epidemic has reduced PWA's access to speech-language therapy (SLT) and limited opportunities for social interaction [8–10]. In this context, tele-lingual therapy (Tele-SLT) has become increasingly important for PWA [11].

Tele-SLT is a method of providing SLT using electronic devices and technology [12,13]. This Tele-SLT assistance can be divided into "assessment" (tele-assessment), which measures and quantitatively translates the patient's condition, and "training" (tele-training), which aims to resolve the patient's difficulties [14]. Furthermore, they can be provided in a synchronous manner, with a speech-language pathologist (SLP) providing SLT in real time via videoconferencing, or in an asynchronous manner, without the presence of an SLP and at one's own pace, or combined types, in which a combination of both is provided [15–17]. The flexible combination of these methods according to the PWA's needs allows for the provision of Tele-SLT tailored to individual requirements. Therefore, the demand for Tele-SLT has increased since the COVID-19 craze [18], but it has not yet reached the same level of diffusion as traditional methods [19,20].

Although the reasons for the struggle with Tele-SLT diffusion are not fully understood, a review of previous studies [15,21–23] suggests that one barrier may be the way information on Tele-SLT for PWA is organized and interpreted. Specifically, it is not well structured in terms of how it is provided and supports the PWA. In other words, there is a lack of clarity about what reliable assessment and training methods exist and how they can be delivered remotely for SLT for PWA. It is undeniable that the absence of reference standards for providing remote SLT has led to situations where it is needed but not provided.

Therefore, this study will use a scoping review approach to investigate Tele-SLT for post-stroke PWA, based on a framework of remote support methods: assessment methods, training methods, and delivery methods, including synchronous, asynchronous, and combined approaches. The significance of this study lies in comprehensively collecting, sorting, and disseminating information that will contribute to the dissemination of Tele-SLT and increase its provision to PWA who have experienced delayed access to professional support since the COVID-19 epidemic.

## Methods

### Study design

Our aim was to systematically identify and analyze the methods (assessment, training) and delivery (synchronous, asynchronous, combined) of Tele-SLT support for post-stroke PWA. Additionally, we sought to identify gaps in the current field of research. To achieve this objective, we used a scoping review methodology [24]. It is important to note that no published articles describe the protocol of this study.

### Information sources and evidence retrieval

This scoping review followed the guidelines of the Preferred Reporting Items for Systematic Reviews and Meta-Analyses extension for Scoping Reviews [25]. We achieved compliance with these guidelines [25]. Searches were conducted in the online databases of Medline [PubMed Interface], Embase, PsycInfo, Cochrane Library, and the ICHUSHI Web (Japan). The search strategy, designed by the expert medical secretary, is detailed in S1 Table. The first search was initiated on March 30, 2023, without specifying the year of publication. No manual search of the grey literature or reference lists of the included articles was performed.

### Selection criteria

We included studies that investigated the assessment or training methods of Tele-SLT for PWA following a stroke. Studies were excluded if they: 1) involved participants under 18 years of age; 2) combined Tele-SLT with in-person SLT; or 3) were review articles (e.g., systematic reviews, scoping reviews, narrative reviews), case reports, qualitative studies, cost-effectiveness analyses, books, conference proceedings, or study protocols. Full details of the inclusion and exclusion criteria are provided in Table 1.

### Screening method

Two authors (YK and SN) independently screened titles and abstracts of identified articles. Subsequently, two other authors (YK and AY) independently reviewed the full texts of the selected articles for inclusion. Any discrepancies between authors during the screening process were resolved through discussion. If a consensus could not be reached, a third author (HN) made the final decision.

**Table 1. Inclusion and Exclusion Criteria.**

| Inclusion criteria | Exclusion criteria |
|---|---|
| 1. Investigated Tele-SLT assessment or training methods for adults with PWA following a stroke. <br> 2. Included participants aged 18 years or older. <br> 3. Were published in English or Japanese. | 1. Included any participants under 18 years of age, or any participants with traumatic brain injury, primary progressive aphasia, dementia, cancer, or conditions other than stroke. <br> 2. Provided in-person SLT as part of the training intervention, except in RCTs, quasi-RCTs where in-person SLT was provided to both the Tele-SLT intervention and control groups. <br> 3. Were published in a language other than English or Japanese. <br> 4. Were review articles (e.g., systematic reviews, meta-analyses, scoping reviews, narrative reviews), case reports, study protocols, qualitative studies, cost-effectiveness analyses, books, or conference proceedings |

Abbreviations: *Tele-SLT* telepractice speech-language therapy, *PWA* people with aphasia, *SLT* speech-language therapy, *RCT* Randomized Controlled Trial

## Data extraction process

Two authors (YK and AY) independently reviewed the full texts of articles selected in the second screening and extracted data. Discrepancies in data extraction were resolved through discussion between the two authors. If consensus could not be reached, a third author (HN) made the final decision. For studies on tele-assessment methods, the following information was collected: author, year of publication, country of the first author, participant characteristics (e.g., sex, age), number of participants, domains covered (e.g., language function, communication, well-being, QOL), software used, electronic devices used, and scale accuracy (e.g., reliability, validity). For studies on tele-training methods, the following information was collected: author, year of publication, study design, participant characteristics (sex, age), number of participants, domains covered (language function, communication, well-being, QOL), software used, electronic devices used, and training outcomes. One author (YK) managed the data using Rayyan reference management software and compiled the data into tables using Excel. Any missing articles were obtained by contacting the corresponding authors.

## Study quality assessment

Two authors (YK, AY) independently assessed the quality of the studies, with the third author (SF) resolving any disagreements. The evaluation checklist [21,26,27], partially adapted by the authors to fit this study's inclusion criteria, consisted of five categories: study design, sample size, demographic variables, aphasia variables (including type classification, severity, and duration since onset), telemedicine characteristics, and data collection methods (specifically, the use of standardized rating scales). Each category was assigned a rating of high, medium, or low (S2 Table). A study was considered to be of high quality if it received no low" ratings and at least four "high" ratings. If a study has no "low" ratings but does not meet the criteria for high quality, it is classified as "medium." Otherwise, it is rated as "low."

## Meta-analysis of the training effects of Tele-SLT

A meta-analysis was conducted using Review Manager 5.4 to evaluate the effects of Tele-SLT training reported in randomized controlled trials (RCTs). The analysis focused on language and communication outcomes. Mean differences (MD), standardized mean differences (SMD) and 95% confidence intervals (CI) were calculated for each outcome using extracted means and standard deviations. Heterogeneity was assessed using $I^2$. A fixed-effects model was used when heterogeneity was not significant ($P > 0.01$), and a random-effects model was used when heterogeneity was significant ($P \leq 0.01$). Outcome measures included, for language function, the Aphasia Quotient (AQ) and its subscales (spontaneous speech, comprehension, naming and repetition) of the Western Aphasia Battery (WAB), WAB-Revised (WAB-R), the Aachen Aphasia Test, and the Boston Naming Test (BNT). For communication function, outcome measures included the Communication Activities Log (CAL), the Communication Effectiveness Index (CETI), and the Communication Abilities in Daily Living (CADL). SMD were calculated when analyzing different outcomes within the same model, whereas MD were calculated for homogenous outcomes. These procedures were based on previous meta-analyses regarding aphasia treatment [28,29].

## Results

### Search results

The initial search yielded a total of 1,484 articles. After removing duplicates, 1,026 articles underwent primary screening based on title and abstract. This screening narrowed the list to 90 articles, which underwent a secondary comprehensive full-text review. Finally, 35 articles

met the inclusion criteria and were included in the current scoping review (Fig 1). S3 Table presents the list of articles included in the secondary screening, the evaluators' (YK, AY) judgments for each, the final decisions, and the reasons for exclusion.

## Trends in selected articles

**Overall trend.** Regarding Tele-SLT support methods, three articles (8.57%) focused on assessment, while 32 (91.43%) focused on training. Regarding delivery methods, 14 articles (40.00%) used synchronous Tele-SLT, 20 (57.14%) used asynchronous Tele-SLT, and 1 (2.86%) used a combined approach. Across all 35 articles, 859 individuals with PWA participated, of whom 693 individuals with chronic PWA were included in 31 articles (88.57%). Study designs for assessment methods were all cross-sectional (n = 3/3). For training methods, designs included pre-post studies (n = 18/32, 56.25%), quasi-randomized controlled trials (quasi-RCTs) (n = 2/32, 6.25%), and RCTs (n = 12/32, 37.50%). All articles were published in English. Tele-SLT support methods, delivery methods, and study designs are summarized in Fig 2.

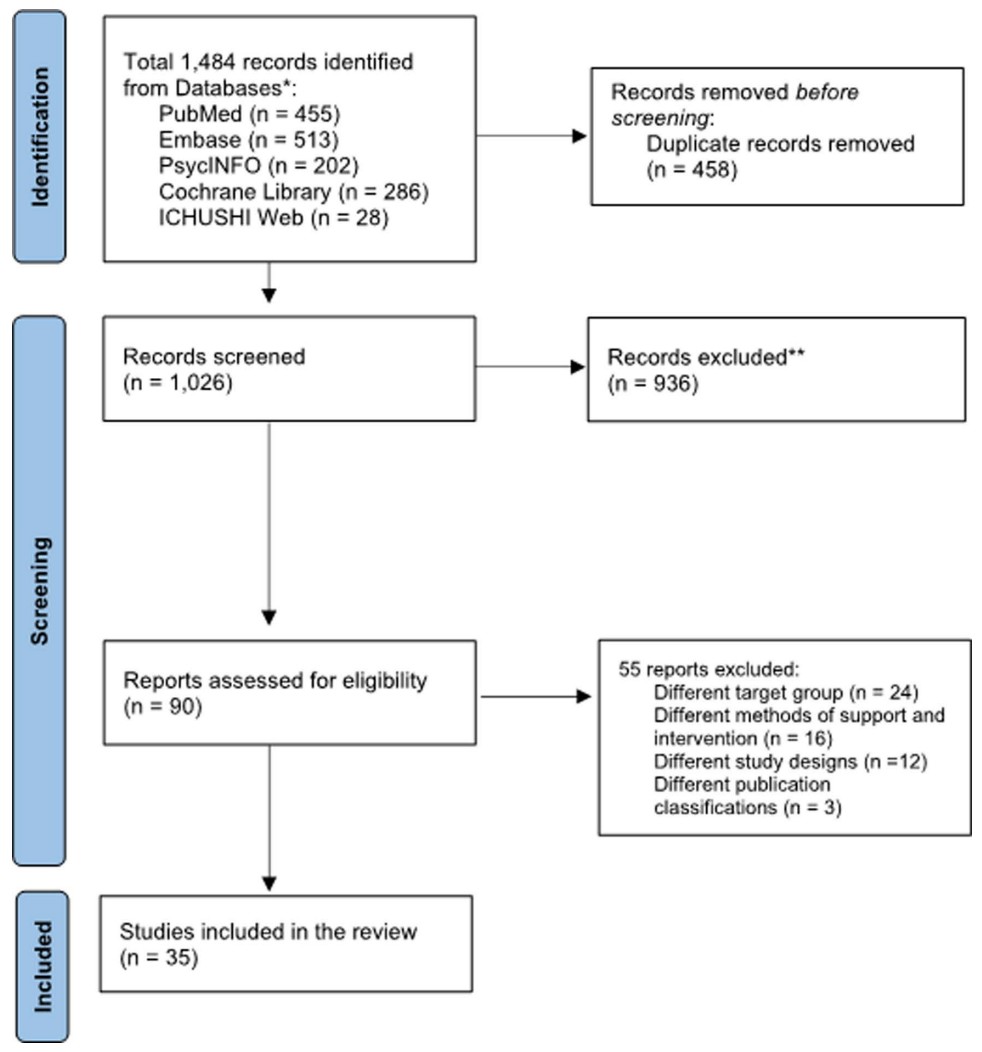

**Fig 1. PRISMA Flow Diagram Illustrating the Selection of Articles Included in the Scoping Review.**

**Trends in research on synchronous Tele-SLT.** Fourteen studies investigated synchronous Tele-SLT, comprising two studies on assessment methods and twelve on training methods (Tables 2 and 3). The assessment studies focused on measuring language function and QOL [30,31]. Training studies targeted language function (n = 8) [32–34,36,39–42], communication (n = 8) [34,39,40,35,37,38,42,43], QOL (n = 7) [34,36,40,37,38,43], and well-being (n = 3) [34,40,43]. Seven studies utilized software specifically designed for PWA. Access2Aphasia was used for assessment [30], while EVA Park [34], Oralys Tele Therapy [35], The Web-based dual card game [36], Rehabilitation Gaming System for aphasia [39,41], and NeuroVR 2.0 [40] were used for training.

Assessment methods were based on the Assessment of Living with Aphasia, the Psycholinguistic Assessments of Language Processing in Aphasia, and the Short Aphasia Test for Gulf Arabic speakers. The psychometric properties examined included intra-rater and inter-rater reliability, criterion validity, and construct validity. Training methods were based on Promoting Aphasics' Communicative Effectiveness [35], Intensive Aphasia Therapy [36,39,41], and conversational training [37,38,40,42,43], with treatment durations ranging from eight days to 24 weeks. Eight studies reported on the maintenance of treatment effects at follow-up periods of 3 to 12 weeks [32–35,39,41–43].

**Trends in research on asynchronous Tele-SLT.** Twenty studies investigated asynchronous Tele-SLT, including one study on assessment methods and nineteen on training methods

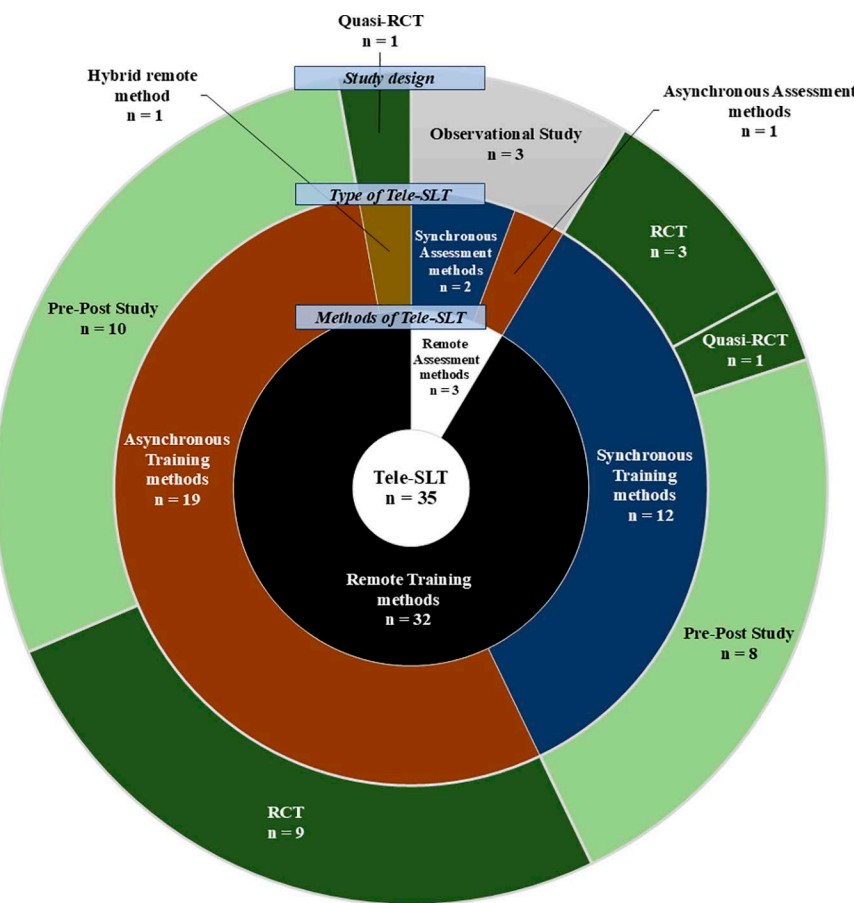

**Fig 2. How to Support and Deliver Tele-SLT and the Design of Studies.**

**Table 2. Details Regarding the Studies on Synchronous Assessment.**

| Author (year, country) | Partici-pants | Number of partic-ipants | Sex of partic-ipants | Mean age of participants (SD) | Assess-ment area | Software | Machin-ery and tools | Results: reliability and validity |
|---|---|---|---|---|---|---|---|---|
| Guo Y.E. et al [30]. (2017, Singapore) | Chronic PWA | 30 | M23, F7 | 61.5 (12.03) | Language function, QOL | Access2Aphasia (software for remote evaluation of ALA and PALPA) | iPad | High correlations with the face-to-face condition, excellent intra-rater and inter-rater reliability were calculated. |
| Altaib, M. K., et al. [31] (2023, Saudi Araibia) | Chronic PWA | 19 | M16, F3 | 48.48 (11.57) | Language function | WebEx (video conferencing system software) | MacBook | There was no significant difference between the results of the Short Aphasia Test for Gulf Arabic speakers when administered remotely and when administered face-to-face, indicating criterion-related validity. |

Abbreviations: *ALA* Assessment for Living with Aphasia, *PALPA* Psycholinguistic Assessments of Language Processing in Aphasia.

Note: We selected "domain of intervention" from "language function," "communication," "well-being," and "quality of life" based on Wallace, S. J., et al. [5].

(Tables 4 and 5). The assessment study focused on measuring language function (n = 1) [44]. Training studies focused on language function (n = 17) [45,46,48–56,58–63], communication (n = 7) [46–48,56,57,59,60], and improving QOL (n = 3) [58,60,62], as well as well-being (n = 1) [58]. Seventeen studies used software specifically designed for PWA. The Mobile Aphasia Screening Test was used for assessment [44]. Training programs utilized various software including Aphasia Scripts [46,47,57,63], Aphasia Mate [48], Oral Reading for Language in Aphasia [49,61], AphasiaRx [51,53], StepByStep aphasia software [50,59], and iAphasia [54].

The assessment study was based on the Korean version of the Frenchay Aphasia Screening Test and examined internal consistency, inter-rater reliability, criterion validity, and construct validity [44]. Training studies employed methods such as word comprehension and production training [45,48–50,52,54–56,58–62] and script training [46,47,51,53,57,63], with intervention durations ranged from single sessions to 26 weeks. Ten studies reported on the maintenance of treatment effects at follow-up periods of 2 to 24 weeks [46,47,50,51,54,55,57–59,61].

**Trends in research on combined Tele-SLT.** The one combined study was a training method study (Table 6). This study focused on improving language function (n = 1) [64] by providing self-designed simultaneous and non-simultaneous training in Face Time and Power Point [64]. Results showed a significant improvement in performance on a unique calling task and confirmed maintenance of calling ability after 6 weeks of training.

## The quality of Tele-SLT studies

Two authors (YK and AY) demonstrated high agreement (97.14%, 34/35) in their assessments of the included studies. Three studies (8.57%) [40,42,62] received a rating of "High," including both synchronous and asynchronous Tele-SLT training interventions. Five studies (14.29%) [31,34,39,59,60] were rated "Moderate," encompassing synchronous Tele-SLT assessment and training, as well as asynchronous Tele-SLT training. The remaining 27 studies (77.14%) were rated as "Low" (Table 7). Notably, "Study Design" (lack of a control group or insufficient sample size in RCTs) and "Data Collection" (insufficient patient outcome measures) were the most frequent reasons for a "Low" rating (n = 19, 18; 54.29%, 51.43%) (Fig 3).

## Training effects of Tele-SLT

We performed a meta-analysis of Tele-SLT training effects using data from four RCTs. Two studies examined synchronous training using the CAL and CETI as out-come measures [39,42], and three examined asynchronous training using the WAB,

**Table 3. Details Regarding the Studies on Synchronous Training.**

**Study design and basic attributes**

| Author (year, country) | Study design | Participants | Number of participants in the intervention group | Number of participants in the control group | Sex of the intervention group | Sex of the control group | Mean age of the intervention group (SD) | Mean age of the control group (SD) |
|---|---|---|---|---|---|---|---|---|
| Agostini, M., et al. [32] (2014, Italy) | Pre-Post study | Chronic PWA | 5 | – | M4, F1 | – | 65.4 (5.7) | – |
| Furnas, D. W., et al. [33] (2014, USA) | Pre-Post study | Chronic PWA | 2 | – | M2 | – | – | – |
| Marshall J., et al. [34] (2016, UK) | Quasi-RCT | Chronic PWA | 10 | 10 | M6, F4 | – | 59.0 (13.61) | 56.6 (9.73) |
| Macoir J. et al. [35] (2017, Canada) | Pre-Post study | Chronic PWA | 20 | – | M14, F6 | – | 63.7 (10.1) | – |
| Pitt, R., et al. [36] (2017, Australia) | Pre-Post study | Chronic PWA | 2 | – | M1, F1 | – | 59.50 (26.16) | – |
| Pitt, R., et al. [37] (2019, Australia) | Pre-Post study | Chronic PWA | 4 | – | M2, F2 | – | 57.50 (16.18) | – |
| Pitt, R., et al. [38] (2019, Australia) | Pre-Post study | Chronic PWA | 18 | – | M9, F9 | – | 58.0 (14.87) | – |
| Grechuta, K., et al. [39] (2019, Spain) | RCT | Chronic PWA | 9 | 8 | M4, F5 | M5, F3 | 55.67 (8.4) | 53.50 (11.33) |
| Giachero A. et al. [40] (2020, Italy) | RCT | Chronic PWA | 18 | 18 | – | – | – | – |
| Grechuta, K., et al. [41] (2020, Spain) | Pre-Post study | Chronic PWA | 10 | – | M5, F5 | – | 57.6 (9.9) | – |
| Øra, H. P., et al. [42] (2020, Norway) | RCT | Convalescent PWA and Chronic PWA | 32 | 30 | M19, F13 | M22, F8 | 64.7 (11.7) | 65.0 (12.2) |
| Cruis, M. et al. [43] (2021, UK) | Pre-Post study | Chronic PWA | 29 | – | M20, F9 | – | 61.3 (11.1) | – |

**Intervention details**

| Author (year, country) | Intervention area | Software | Machinery and tools | Intervention group training content | Control group training content | Frequency of training | Post-therapy results | Maintenance |
|---|---|---|---|---|---|---|---|---|
| Agostini, M., et al. [32] (2014, Italy) | Language function | Skype | Laptop, webcam, headphones | Online picture naming task | Face-to-face naming task for picture words | 8days, 8session | There was a significant improvement in the accuracy of naming words that had been trained in the same way as in person | Three weeks after training, naming test results were maintained. |
| Furnas, D. W., et al. [33] (2014, USA) | Language function | Adobe® Connect, software created independently (software that provides aphasia patients with vocabulary search training) | Desktop computer or laptop | Performed listening comprehension, speech, and typing tasks | – | 8weeks, 3session/week(2hours/session) | Two participants showed improvements in their scores for the items on word production, sentence production, and WAB writing, and one participant showed an improvement in their WAB-AQ score. | After 12 weeks of training, the accuracy of naming words was maintained |
| Marshall J., et al. [34] (2016, UK) | Language function, Communication, Well-being, QOL | EVA Park (VR language training software) | Laptop | Individualized language and communication training, Conversation training | No treatment | 5 weeks, 1h daily | Significant improvement in CADL-2 and CCRSA | Improvement in CADL-2 maintained at 13-week follow-up |

**Table 3.** (Continued)

Intervention details

| Author (year, country) | Intervention area | Software | Machinery and tools | Intervention group training content | Control group training content | Frequency of training | Post-therapy results | Maintenance |
|---|---|---|---|---|---|---|---|---|
| Macoir J. et al. [35] (2017, Canada) | Communication | Oralys TeleTherapy (software specialized for PACE) | Desktop computer | PACE training with 30 individually selected words | – | 3 weeks, 3 sessions/week | Significant improvement in communication effectiveness scores, significant reduction in communication exchange time, significant reduction in the number of communication acts, and significant increase in communication diversity | Improvement in all outcome measures maintained at 6-week follow-up of training |
| Pitt, R., et al [36]. (2017, Australia) | Language function, QOL | The Web-based dual card game, Adobe connect | Desktop computer, webcam, headset microphone | Intensive Constraint-Induced Aphasia Training Implemented training based on the principle of naming | – | 2 weeks, 5 sessions/week (3 hours/session) | The CAT and ALA scores improved. | – |
| Pitt, R., et al. [37] (2019, Australia) | Communication, QOL | Adobe connect | Desktop computer or laptop, webcam, headset microphone | Communication in groups of 2-4 people | – | 12 weeks, 1 session/week (1.5 hours/session) | The CAT and ALA scores improved significantly. | – |
| Pitt, R., et al. [38] (2019, Australia) | Communication, QOL | Adobe connect | Desktop computer or laptop, webcam, headset microphone | Communication in groups of 2-4 people | – | 12 weeks | The ALA, the Quality of Communication Life Scale, the Communication Activity Checklist, and CAT scores significantly improved. | – |
| Grechuta, K., et al. [39] (2019, Spain) | Language function, Communication | Rehabilitation Gaming System for aphasia (software that provides training based on intensive language and movement therapy using virtual reality) | Desktop computer; motion-tracking sensor | PWA members conduct intensive language training and name training in a VR space | Standardized training for specific language disorders | 8 weeks, 5 sessions/week (30-40 minutes/session) | The BDAE scores improved significantly within the treatment group, and the CAL scores improved significantly compared to the control group | After 8 weeks of training, it was confirmed that the results of the BDAE and CAL were maintained |
| Giachero A. et al. [40] (2020, Italy) | Language function, Communication, Well-being, QOL | NeuroVR 2.0 (Software for language and cognitive function training using VR) | Projector, Desktop computer or laptop | Conversation and language function training based on virtual scenarios in VR | Face-to-face conversation training | 24 weeks, 2 sessions/week (2 h/session) | Significant improvement in AAT, C.A.P.P.A. WHOQoL Questionnaire, VASES | – |

*(Continued)*

**Table 3.** (Continued)

Intervention details

| Author (year, country) | Intervention area | Software | Machinery and tools | Intervention group training content | Control group training content | Frequency of training | Post-therapy results | Maintenance |
|---|---|---|---|---|---|---|---|---|
| Grechuta, K., et al. [41] (2020, Spain) | Language function | Rehabilitation Gaming System for aphasia (software that provides training based on intensive language and movement therapy using virtual reality) | Desktop computer, VR headset, motion-tracking sensor | PWA members conduct intensive language training and name training in a VR space | – | 8weeks, 5sessions/ week (30–40 minutes/ session) | The Vocabulary Test scores significantly improved during and after training | After 8 weeks of training, the results of The Vocabulary Test were significantly improved compared to before training |
| Øra, H. P., et al. [42] (2020, Norway) | Language function, Communication | Cisco Jabber/Acano (video conferencing software) | Laptop, webcam, external speakers | Practice word naming, reading comprehension, and communication tailored to individual needs | Standard Care | 8weeks, 5days/ week(60minutes/ session) | The scores on the sub-tests of the Norwegian Basic Aphasia Assessment and the CETI improved significantly | After 12 weeks of training, the results of the subtests of the Norwegian Basic Aphasia Assessment improved compared to the control group, while the results of the CETI test were maintained |
| Cruis, M. et al. [43] (2021, UK) | Communication, QOL, Well-being | Skype | Desktop computer or laptop, iPad | Instruction on the use of equipment and conversation training, as well as practice of communication skills in this context | – | 8 weeks, twice a week (1h/ session) | Social network assessment showed an increase in the number of social contacts and a significant improvement in CCRSA and ALA | Social network coverage, CCRSA, and ALA maintained at 8-week follow-up |

Abbreviations: *WAB* the Western Aphasia Battery, *WAB-AQ* the Western Aphasia Battery Aphasia Quotient, *CADL-2* Communication ADL Test-2, *CCRSA* Communication Confidence Rating Scale for Aphasia, *PACE* Patient-Accessible Communication Environment, *CAT* Comprehensive Aphasia Test, *ALA* Assessment for Living with Aphasia, *BDAE* Boston Diagnostic Aphasia Examination, *CAL* Communicative Activity Log, *AAT* Aachen Aphasia Test, *C.A.P.P.A* Conversation Analysis Profile For People with Aphasia, *WHOQOL* The World Health Organization Quality of Life, *VASES* Visual Analog Self-Esteem Scale, *CETI* Communicative Effectiveness Index.

Note: We selected "domain of intervention" from "language function," "communication," "well-being," and "quality of life" based on Wallace, S. J., et al. [5].

Table 4. Details of the Studies of Asynchronous Assessment.

| Author (year, country) | Participants | Number of participants | Sex of participants | Mean age of participants (SD) | Assessment area | Software | Machinery and tools | Results: reliability and validity |
|---|---|---|---|---|---|---|---|---|
| Choi, Y. H. et al. [44] (2015, Korea) | Chronic PWA | 30 | M25, F5 | 53.67 (5.27) | Language function | MAST (software version of FAST) | iPad | Inter-rater reliability, internal consistency reliability, and convergent validity (correlation with K-FAST and K-WAB-AQ) were good. Sensitivity and specificity for determining the presence of aphasia were also good. |

Abbreviations: *K-WAB-AQ* Korean version the Western Aphasia Battery Aphasia Quotient, *MAST* Mobile Aphasia Screening Test, *K-FAST* Korean version Frenchay Aphasia Screening Test.

Note: We selected "domain of intervention" from "language function," "communication," "well-being," and "quality of life" based on Wallace, S. J., et al. [5].

WAB-R as the outcome measure [45,56,62]. For synchronous training, there was no significant difference between the intervention and control groups in combined CAL and CETI scores (fixed-effects model, SMD = -0.10, 95% CI = -0.57 to 0.37, $P > 0.05$) (Fig 4). For asynchronous training, analysis of the WAB-AQ showed no significant difference between groups (fixed-effects model, SMD = 0.24, 95% CI = -0.17 to 0.65, $P > 0.05$) (Fig 5). However, a significant difference was found in the combined WAB subscales (Spontaneous speech, Comprehension, Repetition, Naming) (fixed-effects model, SMD = 0.23, 95% CI = 0.02 to 0.44, $P < 0.05$). Despite the significant result for the combined subscales, no significant differences were observed for individual subscales: Spontaneous speech (fixed-effects model, SMD = 0.20, 95% CI = -0.26 to 0.67, $P > 0.05$), Comprehension (fixed-effects model, SMD = 0.28, 95% CI = -0.13 to 0.69, $P > 0.05$), Repetition (fixed-effects model, SMD = 0.13, 95% CI = -0.28 to 0.54, $P > 0.05$) and Naming (fixed-effects model, SMD = 0.31, 95% CI = -0.11 to 0.72, $P > 0.05$) (Fig 6)

## Discussion

### Findings of this Tele-SLT review

This review investigated Tele-SLT support and delivery methods for PWA. Our findings reveal a growing body of research examining the use of specialized software for Tele-SLT, particularly for individuals with chronic PWA; many studies demonstrate its benefits. However, three key challenges emerged. First, research on tele-assessment is scarce compared to research on tele-training. Second, there is insufficient research on synchronous remote SLT compared to asynchronous remote SLT, and RCTs have yet to demonstrate a significant training effect for synchronous interventions compared to asynchronous interventions. Third, the overall methodological quality of Tele-SLT research is low. These findings suggest that the availability of diverse Tele-SLT approaches may be limited, and the efficacy of Tele-SLT, particularly synchronous training, warrants further investigation. As the demand for Tele-SLT increases, addressing these challenges is crucial.

### Bias in the number of studies on the content of support in Tele-SLT

Only three studies (8.57%) involving tele-assessments were included in this study, highlighting the challenges associated with conducting tele-assessments. Specifically, it may be difficult to remotely determine whether an individual is PWA and to obtain detailed information about them, potentially hindering the provision of tele-training [65]. A significant challenge is the influence of digital device functionality and the environment on the interpretation of PWA

**Table 5. Details Regarding the Studies on Asynchronous Training.**

**Study design and basic attributes**

| Author (year, country) | Study design | Participants | Number of participants in the intervention group | Number of participants in the control group | Sex of the intervention group | Sex of the control group | Mean age of the intervention group (SD) | Mean age of the control group (SD) |
|---|---|---|---|---|---|---|---|---|
| Kats, R. C. et al. [45] (1997, USA) | RCT | Chronic PWA | 21 | 34 (dummy stimulus 19, no treatment 15) | - | - | 61.6 (10.0) | Dummy stimulus: 66.4 (6.0), No treatment: 62.8 (5.1) |
| Cherney, L. R., et al. [46] (2008, USA) | Pre-Post study | Chronic PWA | 3 | - | M2, F1 | - | 64.0 (12.77) | - |
| Manheim, L.M. et al. [47] (2009, USA) | Pre-Post study | Chronic PWA | 20 | - | M13, F7 | - | 54.80 (15.25) | - |
| Archibald, L. M. D., et al. [48] (2009, Canada) | Pre-Post study | Chronic PWA | 8 | - | M6, F2 | - | 71.13 (11.12) | - |
| Cherney, L. R [49]. (2010, USA) | RCT | Chronic PWA | 11 | 14 | M 8, F 3 | M8, F6 | 56.6 (9.2) | 61.1 (14.8) |
| Palmer, R., et al. [50] (2012, UK) | RCT | Chronic PWA | 16 | 17 | M9, F7 | M12, F5 | 69.5 (12.2) | 66.2 (12.3) |
| van Vuuren, S., et al. [51] (2014, USA) | Pre-Post study | Chronic PWA | 8 | - | - | - | 52.0 (14.0) | - |
| Kurland, J., et al. [52] (2014, USA) | Pre-Post study | Chronic PWA | 5 | - | M2, F3 | - | 67.6 (8.26) | - |
| Cherney, L.R., et al. [53] (2015, USA) | Pre-Post study | Chronic PWA | 8 | - | M6, F2 | - | 52.0 (14.0) | - |
| Choi, Y. H., et al. [54] (2016, Korea) | Pre-Post study | Chronic PWA | 8 | - | M4, F4 | - | 50.75 (8.88) | - |
| Kurland, J. et al. [55] (2018, USA) | Pre-Post study | Chronic PWA | 21 | - | M13, F8 | - | 66.4 (8.4) | - |
| Zhou, Q., et al. [56] (2018, China) | RCT | Convalescent PWA | 10 | 10 | M7, F3 | M6, F4 | 59.80 (11.26) | 56.50 (14.34) |
| Cherney, L.R. et al. [57] (2019, USA) | Pre-Post study | Chronic PWA | 20 | - | M14, F6 | - | 56.9 (8.4) | - |
| Maresca, G. et al. [58] (2019, Italy) | RCT | Convalescent PWA | 15 | 15 | M7, F8 | M7, F8 | 51.1 (10.4) | 51.4 (12.7) |
| Palmer, R., et al. [59] (2019, UK) | RCT | Chronic PWA | 83 | 86 | M47, F36 | M54, F32 | 64.9 (13.0) | Control A: 64.9 (13.0), Control B: 63.8 (13.1) |
| Spaccavento, S. et al. [60] (2021, Italy) | RCT | Acute PWA | 13 | 9 | M9, F4 | M7, F2 | 57.38 (9.23) | 64.11 (15.04) |
| Cherney, L.R. et al. [61] (2021, USA) | RCT | Chronic PWA | 19 | 13 | M10, F9 | M9, F4 | 58.27 (13.55) | 55.19 (11.46) |
| Braley M., et al. [62] (2021, Italy) | RCT | Convalescent PWA | 17 | 15 | M10, F7 | M8, F7 | 59.8 (10.0) | 64.2 (9.9) |
| Cherney, L.R., et al. [63] (2022, USA) | Pre-Post study | Chronic PWA | 16 | - | M10, F6 | - | 61.54 (12.96) | - |

**Intervention details**

| Author (year, country) | Intervention area | Software | Machinery and tools | Intervention group training content | Control group training content | Frequency of training | Post-therapy results | Maintenance |
|---|---|---|---|---|---|---|---|---|
| Kats, R. C. et al. [45] (1997, USA) | Language function | Digital program for visual matching and reading (developed for research) | Desktop personal computer | Word and Sentence Reading Assignment | Non-verbal, cognitive rehabilitation software, computer games, no treatment | 3h per week for 26 weeks | PICA and WAB improved significantly | - |

*(Continued)*

Table 5. (Continued)

Intervention details

| Author (year, country) | Intervention area | Software | Machinery and tools | Intervention group training content | Control group training content | Frequency of training | Post-therapy results | Maintenance |
|---|---|---|---|---|---|---|---|---|
| Cherney, L. R., et al. [46] (2008, USA) | Language function, Communication | AphasiaScripts™ (software that uses animated agents that function as virtual therapists to support skript traning) | Desktop computer or laptop | Script training with a virtual therapist | – | 9 weeks, 7 sessions/week (30minetes/session) | Two out of three participants showed improvements in content, grammar, and word production speed in two of the three scripts. One participant showed improvements in WAB-AQ and CETI scores | After 6 weeks of training, the WAB-AQ and CETI scores improved and were maintained |
| Manheim, L.M. et al. [47] (2009, USA) | Communication | Aphasia Scripts™ (computer-based script training program) | Laptop | Script training with a virtual therapist | – | 9 weeks, 30 min daily | BOSS-CD decreased (decreased sense of communication difficulty) | BOSS-CD values were maintained at the 6-week follow-up |
| Archibald, L. M. D., et al. [48] (2009, Canada) | Language function, Communication | AphasiaMate | Desktop computer or laptop | Language function tasks such as auditory comprehension, reading comprehension, mean-ing therapy, and calculation | – | 15 weeks, at least once a week (60 minutes or more per session) | Significant improvement in the results of the lower-level tests for WAB and ASHA-FACS | – |
| Cherney, L. R [49]. (2010, USA) | Language function | Oral Reading for Language in Aphasia (ORLA) | Desktop computer or laptop | Comprehension of sentences by software, recitation of words, training on reading words aloud | Sentence comprehension, word recitation, word reading practice | 6–22 weeks, 1–4 sessions/week (1h/session) | More patients in the treatment group had improved WAB-AQ, and no significant difference in WAB-AQ change was noted between the treatment and control groups. | – |
| Palmer, R., et al. [50] (2012, UK) | Language function | The StepByStep aphasia software (software for aphasia training) | Desktop computer or laptop | Word Search Task | Participation in communication support groups, and everyday activities such as conversation, reading and writing | 20 weeks, 3 sessions/week (20minutes/session) | The accuracy of word naming in the Object and Action Naming Battery has significantly improved | After 12 weeks of training, the accuracy of naming lifesaving words is maintained |

(Continued)

**Table 5.** (Continued)

Intervention details

| Author (year, country) | Intervention area | Software | Machinery and tools | Intervention group training content | Control group training content | Frequency of training | Post-therapy results | Maintenance |
|---|---|---|---|---|---|---|---|---|
| van Vuuren, S., et al. [51] (2014, USA) | Language function | AphasiaRx™ (Software for language function training using script training methods) | Laptop | Script training with a virtual therapist | – | 3 weeks, 6 sessions/week (up to 90minutes/session) | The accuracy of the words used in the 10 sentences practiced improved significantly compared to before training | After 3, 6, and 12 weeks of training, the accuracy of naming words in the trained script is maintained |
| Kurland, J., et al. [52] (2014, USA) | Language function | iBooks (with a playback function for videos such as audio comprehension of words edited in imove and pronunciation hints) | iPad | Vocabulary reading comprehension, repetition, and naming practice | – | 24 weeks, 5–6 sessions/week(20minutes/session) | 4–5 participants improve the accuracy of the expression of not only trained words but also untrained words | – |
| Cherney, L.R., et al. [53] (2015, USA) | Language function | AphasiaRx™ (Software for language function training using script training methods) | Laptop | Script training with a virtual therapist | – | 3 weeks, 6 sessions/week (90 minutes/session) | Significant improvement in the accuracy of word naming in scripts | – |
| Choi, Y. H., et al. [54] (2016, Korea) | Language function | iAphasia (language training software) | iPad | Training tasks for auditory comprehension, reading comprehension, repetition, memorization, writing, and language fluency | – | 4 weeks | Significant improvement in the lower ratings of WAB-AQ and WAB | Improvement in WAB-AQ maintained after 4 weeks of training |
| Kurland, J. et al. [55] (2018, USA) | Language function | iBooks | iPad | Training in calling pictures (objects and actions) presented in software | – | 5–6 sessions a week for 24 weeks (20 min/session) | Significant improvements in calling performance for used and unused words during training | Maintain nominal performance at 16 weeks follow-up |
| Zhou, Q., et al. [56] (2018, China) | Language function, Communication | Language training software for people with aphasia from The Wispirit Inc. (66nao.com) | Desktop computer or laptop or Tablet device | Listening comprehension, reading comprehension, repetition, dictation, and writing practice exercises | Communication about family topics | 4 weeks, 7 sessions/week (30 minutes/session) | Significant improvement in WAB and CADL scores | – |

*(Continued)*

**Table 5.** (Continued)

Intervention details

| Author (year, country) | Intervention area | Software | Machinery and tools | Intervention group training content | Control group training content | Frequency of training | Post-therapy results | Maintenance |
|---|---|---|---|---|---|---|---|---|
| Cherney, L.R. et al. [57] (2019, USA) | Communication | Aphasia Scripts - VR | Desktop computer or laptop | Script training with a virtual therapist | – | 1 time only, 60 min. | Training scripts were evaluated by NORLA-6 and showed significant improvement | These improvements were maintained for after 2 weeks. |
| Maresca, G. et al. [58] (2019, Italy) | Language function, Well-being, QOL | Virtual Reality Rehabilitation System (VRRS-Tablet) | Tablet device | Object nomenclature training, sentence construction, sentence composition nomenclature training | Traditional speech therapy | 12 weeks, 5 sessions/week (50 min/session) | Significant improvements in TT, ADRS, ENPA EQ-5D, and PIADS | Improvement was still observed in all outcomes after 12 weeks of training |
| Palmer, R., et al. [59] (2019, UK) | Language function, Communication | The StepByStep aphasia software (software for aphasia training) | Desktop computer or laptop or Tablet device | Search task for words related to PWA | Paper-based cognitive function activities (Sudoku, Spot the Difference, etc.) | 24 weeks, 7 sessions/week (20–30 minutes/session) | Significant improvement in the results of the 100-word picture naming test | After 12 and 24 weeks of training, the results for word recognition were maintained |
| Spaccavento, S. et al. [60] (2021, Italy) | Language function, Communication, QOL | Software programs edited by Erickson (Edizioni Centro Studi Erickson, S.p.A.) for rehabilitation language skills training | Desktop computer or laptop | Software training for word and sentence expression and language comprehension | Therapist training in word and sentence expression and language comprehension | 8 weeks, 5 sessions/week (50 min/session) | Significant improvement in AAT, FOQ-A, FAM, and QLQA | – |
| Cherney, L.R. et al. [61] (2021, USA) | Language function | Web-based Oral Reading for Language in Aphasia (ORLA® Web version) | Laptop, audio headset, and webcam | Listening comprehension, reading comprehension, recitation, and oral reading tasks using a virtual avatar of the therapist | PopCap's commercial game Bejeweled 2©, which does not target language skills | 6 weeks, 6 sessions/week (90 min/session) | Significant improvement in WAB-R-LQ | Further improvement was observed at the 6-week follow-up |

*(Continued)*

**Table 5.** (Continued)

Intervention details

| Author (year, country) | Intervention area | Software | Machinery and tools | Intervention group training content | Control group training content | Frequency of training | Post-therapy results | Maintenance |
|---|---|---|---|---|---|---|---|---|
| Braley M., et al. [62] (2021, Italy) | Language function, QOL | Constant Therapy-Research (software that provides systematic and structured therapy like that usually provided by speech-language-hearing therapists) | Tablet device | Training that spans the cognitive, speech, and language domains | Independent practice using a aphasia treatment workbook | 8 weeks, 5 sessions/week (30 minutes/session) | Significant improvement in WAB-AQ | – |
| Cherney, L.R., et al. [63] (2022, USA) | Language function | AphasiaScripts™ (software program that uses animated agents to function as virtual therapists to support aphasia patients' speech training) | Laptop | Script training with a virtual therapist | – | 3 weeks, 6 sessions/week (30 minutes/session) | Significant increase in the accuracy of words in the trained script and the number of words per minute | – |

Abbreviations: *PICA* Porch Index of Communicative Ability, *WAB* the Western Aphasia Battery, *WAB-AQ* the Western Aphasia Battery Aphasia Quotient, *CETI* Communicative Effectiveness Index, *BOSS* Burden of Stroke Scale, *CD* Communication Difficulty, *ASHA-FACS* American Speech-Language-Hearing Association Functional Assessment of Communication Skills for Adults, *CADL* Communication ADL Test, *NORLA-6* Naming and Oral Reading for Language in Aphasia 6-Point Scale, *TT* Token Test, *ADRS* Aphasic Depression Rating Scale, *ENPA* Esame Neurologico Per l'Afasia, *EQ-5D* Euro-Qol-5D, *PIADS* Psychosocial Impact of Assistive Devices Scale, *AAT* Aachen Aphasia Test, *I-FOQ-A* Italian Version of Functional Outcome Questionnaire for Aphasia, *FAM* Functional Assessment Measure, *QLQA* Quality of Life Questionnaire for Aphasics, *WAB-R -LQ* Western Aphasia Battery-Revised Language Quotient.

Note: We selected "domain of intervention" from "language function," "communication," "well-being," and "quality of life" based on Wallace, S.J., et al. [5].

**Table 6. Details Regarding the Study on Combined Training.**

**Study design and basic attributes**

| Author (year, country) | Study design | Partici-pants | Number of participants in the intervention group | Number of participants in the control group | Sex of the interven-tion group | Sex of the con-trol group | Mean age of the intervention group (SD) | Mean age of the control group (SD) |
|---|---|---|---|---|---|---|---|---|
| Woolf, C., et al. [64] (2016, UK) | Quasi-RCT | Chronic PWA | 10 | 10 | Remote from univer-sity: M 3, F 2, Remote from clinical site: M 4, F 1 | Face-to-face: M 3, F 2, Attention control: M 4, F 1 | Remote from uni-versity: 67.2 (6.98), Remote from clinical site: 58.6 (14.38) | Face-to-face: 57.8 (15.14), Attention con-trol: 53.0 (13.93) |

**Intervention details**

| Author (year, country) | Inter-vention area | Software | Machinery and tools | Intervention group training content | Control group train-ing content | Frequency of training | Post-therapy results | Maintenance |
|---|---|---|---|---|---|---|---|---|
| Woolf, C., et al. [64] (2016, UK) | Lan-guage function | FaceTime, Power Point | iPad | Training in naming and understanding words, both synchronous and asynchronous | Comprehension, call training, simulta-neous conversation training | 4 weeks, twice a week (1 h/ session) | Significant improve-ments in name calling performance for 100 items | Nominal results maintained at the 6-week follow-up |

Note: We selected "domain of intervention" from "language function," "communication," "well-being," and "quality of life" based on Wallace, S.J., et al. [5].

outcomes. For instance, in videoconferencing communication assessments, there is a concern that microphones may fail to capture unintelligible or faint utterances caused by speech or language disorders, making it challenging to derive clinical insights into spoken language [66]. Furthermore, loss of video data has been noted due to volume changes associated with the PWA's sitting position and inadequate computer capacity, even when the microphone's audio input is normal [66]. Therefore, implementing tele-assessments may necessitate selecting an assessment modality based on the severity of the PWA's aphasia and the device's capabilities, as well as verifying the assessment environment in advance [66]. From clinical or research perspectives, establishing such protocols and checklists is essential [67]. In the context of Tele-SLT, utilizing conversation recording protocols from Aphasiabank, an online platform for sharing PWA data, may be beneficial [68]. If these procedures are made accessible to SLP, other professionals, and PWA in a format compatible with Tele-SLT, it is likely that tele-evaluation will gain popularity as a user-friendly approach.

## Limited research on synchronous Tele-SLT delivery

Our review found that only 14 of the 35 included studies (40.00%) investigated synchronous Tele-SLT, suggesting potential challenges in its implementation. This scarcity may indicate a lack of strategies for addressing communication, QOL, and well-being in PWA through syn-chronous Tele-SLT [69–71]. Such support is often associated with, and facilitated by, the inter-active communication inherent in synchronous interventions. Interactive communication promotes social participation in PWA [72], which is directly linked to QOL and well-being [5]. Therefore, synchronous Tele-SLT may be crucial for supporting social participation and QOL in PWA, and addressing barriers to its implementation is essential.

Potential barriers may include concerns about privacy and confidentiality, as well as a lack of ethical and technical support for using video conferencing in synchronous Tele-SLT [73]. Developing guidelines and checklists regarding privacy, confidentiality, and the use of communication technology for PWA could help mitigate these issues [66]. Without standard-ized practices for video conferencing in SLT, the expansion of synchronous Tele-SLT may be hindered, and the inconclusive training effects suggested by our meta-analysis may remain

**Table 7. Assessing the Quality of Studies.**

| Tele-SLT | Author, Year of publication | Study Design | Demographic variables | Aphasia variables | Telehealth characteristics | Data collection | Overall rating |
|---|---|---|---|---|---|---|---|
| **Assessment methods** | Choi, Y.H., et al., 2015 [30] | High | Low | High | High | Low | Low |
| | Guo, Y.E., et al., 2017 [44] | Low | High | High | High | Moderate | Low |
| | Altaib, M. K., et al., 2023 [31] | Moderate | High | High | High | Moderate | Moderate |
| **Synchronous training methods** | Agostini, M., et al., 2014 [32] | Low | High | Moderate | High | Low | Low |
| | Furnas, D. W., et al., 2014 [33] | Low | High | High | Moderate | Low | Low |
| | Marshall, J., et al., 2016 [34] | High | Moderate | Moderate | Moderate | Moderate | Moderate |
| | Macoir, J., et al., 2017 [35] | Low | Low | Moderate | High | Low | Low |
| | Pitt, R., et al., 2017 [36] | Low | Moderate | Moderate | High | High | Low |
| | Pitt, R., et al., 2019 [37] | Low | Moderate | High | High | High | Low |
| | Pitt, R., et al., 2019 [38] | Low | Low | Low | High | High | Low |
| | Grechuta, K., et al., 2019 [39] | Moderate | Moderate | High | High | Moderate | Moderate |
| | Giachero, A., et al., 2020 [40] | High | High | Moderate | High | High | High |
| | Grechuta, K., et al., 2020 [41] | Low | Moderate | High | High | Moderate | Low |
| | Øra, H. P., et al., 2020 [42] | High | High | Moderate | High | High | High |
| | Cruice, M., et al., 2021 [43] | Low | Low | Moderate | High | Moderate | Low |
| **Asynchronous training methods** | Ksts, R. C., et al., 1997 [45] | High | High | Moderate | Low | Low | Low |
| | Cherney, L. R., 2008 [46] | Low | Moderate | High | Moderate | Moderate | Low |
| | Manheim, L. M., 2009 [47] | Low | High | Moderate | Moderate | Low | Low |
| | Archibald, L. M. D., et al., 2009 [48] | Low | High | High | High | Moderate | Low |
| | Cherney, L.R., 2010 [49] | High | Low | High | Moderate | Low | Low |
| | Palmer, R., et al., 2012 [50] | High | Low | High | Moderate | Moderate | Low |
| | van Vuuren, S., et al., 2014 [51] | Low | Low | Moderate | Moderate | Low | Low |
| | Kurland, J., et al., 2014 [52] | Low | Moderate | High | High | Low | Low |
| | Cherney, L.R., et al., 2015 [53] | Low | High | High | Moderate | Low | Low |
| | Choi, Y. H., et al., 2016 [54] | Low | High | High | Moderate | Low | Low |
| | Kurland, J., et al., 2018 [55] | Low | High | High | Moderate | Low | Low |
| | Zhou, Q., et al., 2018 [56] | High | Low | High | Moderate | Low | Low |
| | Cherney, L.R., et al., 2019 [57] | Low | High | High | Moderate | Low | Low |
| | Maresca, G., et al., 2019 [58] | High | High | Low | High | Low | Low |
| | Palmer, R., et al., 2019 [59] | High | Moderate | High | Moderate | High | Moderate |
| | Spaccavento, S., et al., 2021 [60] | Moderate | Moderate | Moderate | High | Moderate | Moderate |
| | Cherney, L. R., et al., 2021 [61] | High | High | High | High | Low | Low |
| | Braley, M., et al., 2021 [62] | High | High | High | High | Moderate | High |
| | Cherney, L. R., et al., 2022 [63] | Low | High | High | Moderate | Low | Low |
| **Combined Training method** | Woolf, C., et al., 2015 [64] | Moderate | Low | Low | High | Low | Low |

Note: Bold indicates high assessment, italic indicates low assessment, and underline indicates moderate assessment. This research quality assessment tool is based on Teti et al (2023) [21] and Salis et al. (2021) [26] draws upon the information from the McMaster University Health Evidence Quality Assessment Tool Dictionary. For a study to receive an overall high rating (excluding low ratings), it must score highly on at least four out of the five criteria.

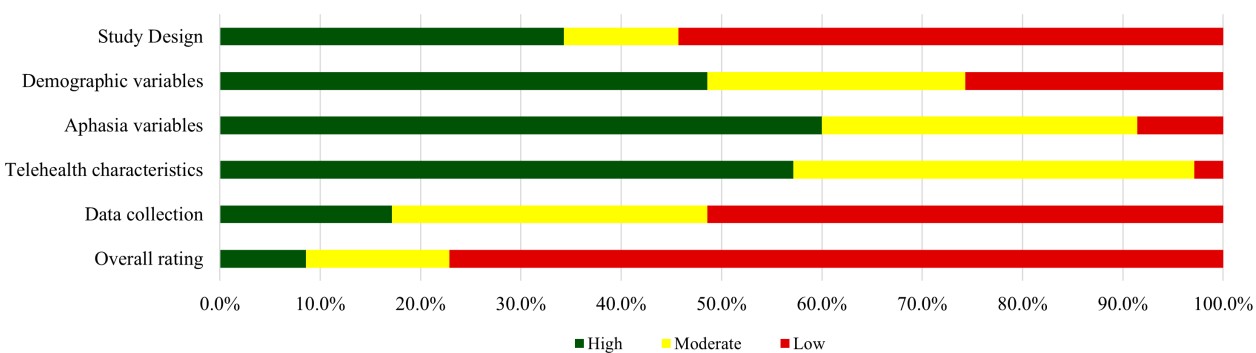

**Fig 3. Summarize the Results of the Study Quality Assessment.**

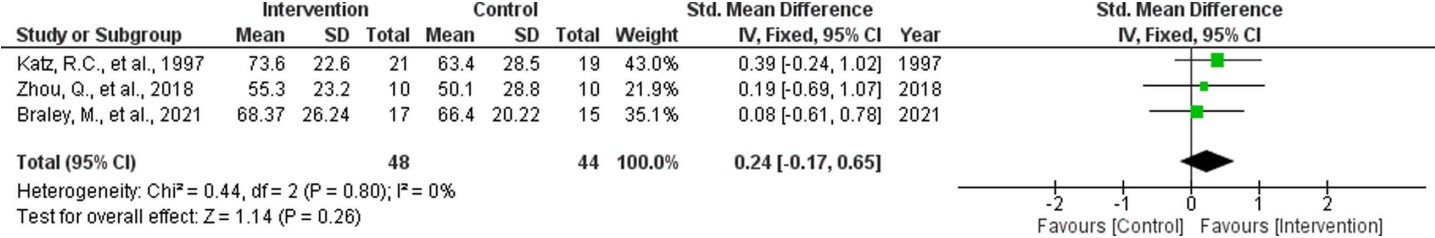

**Fig 4. Results of a Meta-Analysis of the Effects of Synchronous Training on Communication Outcomes.**

**Fig 5. Results of a Meta-Analysis of the Effects of Asynchronous Training on the Western Aphasia Battery-Aphasia Quotient.**

unresolved. Without addressing these challenges, we risk missing opportunities to provide effective SLT to PWA who have faced increased difficulties with social participation and daily living since the onset of the COVID-19 pandemic.

## Quality of Tele-SLT studies

The quality of Tele-SLT studies has been consistently rated as low. The main reasons for this can be summarized as follows: poor baseline data due to the lack of standardized assessment methods for PWA, as noted in the 'Data Collection' section, and an undefined target group and insufficient sample size, as mentioned in the 'Study Design' section. Regarding the former, the shortcomings of standardized assessment methods in SLT have been documented [74], and adapting traditional tests for Tele-SLT, as has been done with the WAB [75], may be necessary. With regard to the latter, barriers such as inattention and poor auditory comprehension in PWA have hindered their participation in telemedicine [66]. This is why maintaining and increasing sample sizes in studies involving PWA has been challenging. Although

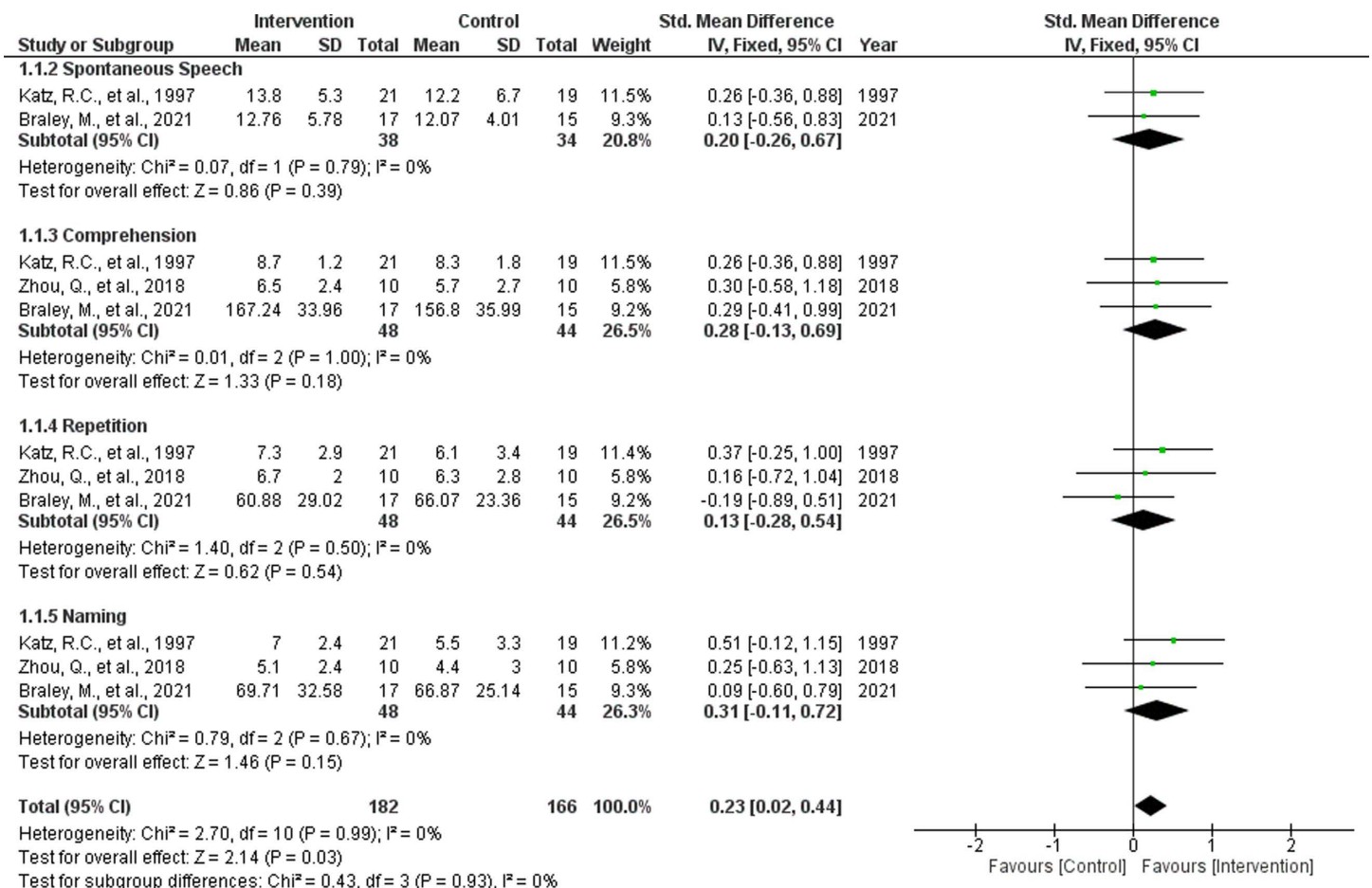

**Fig 6. Results of a Meta-Analysis of the Effects of Asynchronous Training on the Western Aphasia Battery Subscales, Western Aphasia Battery Revised Subscales.**

some suggest that patient sampling is easier in telerehabilitation [76], this may not hold true for PWA. Therefore, study designs must consider the cognitive and language functions of PWA to ensure an adequate sample size.

## Future directions in the post-COVID-19 era

The recent COVID-19 pandemic has increased the challenges of delivering in-person SLT, which has led to a surge in global interest in Tele-SLT. Establishing effective tele-assessment and synchronous training methods, which our review identified as areas in need of further development, is a high priority. Widespread implementation of Tele-SLT has the potential to expand access to SLT for a greater number of PWA. To achieve this, it is crucial to determine which types of Tele-SLT (assessment, training, synchronous, asynchronous, or combined) are most effective for specific PWA profiles. Therefore, improving the quantity and quality of research across all areas of Tele-SLT, including the development of tele-assessment tools and synchronous training protocols specifically designed for PWA, is essential.

One approach to address this challenge is to promote the development and use of specialized software for PWA. Such software was utilized in 68.57% (n = 24/35) of the studies included in this review. Specialized software can facilitate the management of patient information in rehabilitation settings and, under appropriate conditions, enable high-quality audio

and video transmission [19]. Furthermore, integrating software into Tele-SLT practice is thought to reduce the burden on speech-language pathologists and promote wider adoption of Tele-SLT [77]. Therefore, developing and disseminating PWA-specific software could be instrumental in expanding access to and strengthening the evidence base for Tele-SLT.

## Limitations of this review

This review has five main limitations. First, the failure to use specific checklists for the quantitative quality assessment of the included studies may introduce uncertainty regarding the consistency of the study quality in this scoping review design. Second, while a comprehensive literature search was conducted, the exclusion of articles in languages other than English and Japanese, as well as gray literature and manual searches of specific databases, may have led to selection bias. Third, the exclusion of articles in languages other than English and Japanese in the meta-analysis limited the number of studies included for analysis, and we could not conduct subgroup analyses based on participant characteristics or details of interventions. Therefore, caution is needed when interpreting the results of the meta-analysis. Fourth, as we focused on studies that included multiple participants with aphasia, single-case study designs were excluded. Finally, as with all database searches, it is impossible to completely eliminate publication bias [78].

## Conclusion

In this study, we reviewed assessment and training methods in Tele-SLT for PWA and categorized them into synchronous, asynchronous, and combined methods. Despite an initial search of 1,484 articles from five databases, only 35 met our strict inclusion criteria. This finding underscores the scarcity of research on tele-assessment and synchronous training and highlights the overall poor quality of available studies. Given the increasing societal involvement of PWA, research in these areas is particularly critical in the post-COVID-19 era, and continued research is necessary. However, all studies showed valid results, and Tele-SLT methodologies, including digital programs dedicated to PWA, appear to be indicators of future solutions to the problem.

## Supporting information

**S1 Table. Search formula used in this study.**
(DOCX)

**S2 Table. Criteria for evaluating the quality of studies.**
(DOCX)

**S3 Table. Results by evaluator and final evaluation of the secondary screening inclusion articles.**
(DOCX)

## Acknowledgments

The authors extend their sincere gratitude to Ms. Satomi Kojima, a medical librarian, for her invaluable assistance in formulating the search strategy for this paper.

## Author contributions

**Conceptualization:** Yuhei Kodani, Shinsuke Nagami, Ayaka Yokozeki.

**Data curation:** Yuhei Kodani, Ayaka Yokozeki.

**Investigation:** Yuhei Kodani, Ayaka Yokozeki, Shinya Fukunaga, Hikaru Nakamura.

**Project administration:** Yuhei Kodani, Shinsuke Nagami.

**Writing – original draft:** Yuhei Kodani, Shinsuke Nagami.

**Writing – review & editing:** Yuhei Kodani, Shinsuke Nagami, Ayaka Yokozeki, Shinya Fukunaga, Katsuya Nakamura, Hikaru Nakamura.

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
