## [Decision Letter · Decision Letter 0]

8 Oct 2024

PONE-D-24-13841Current status of tele-speech language therapy by type and support for patients with post-stroke aphasia: a scoping reviewPLOS ONE

Dear Dr. Nagami,

Thank you for submitting your manuscript to PLOS ONE. After careful consideration, we feel that it has merit but does not fully meet PLOS ONE’s publication criteria as it currently stands. Therefore, we invite you to submit a revised version of the manuscript that addresses the points raised during the review process.

We look forward to receiving your revised manuscript.

Kind regards,

Josep Vidal-Alaball, MD, PdD, MPH

Academic Editor

PLOS ONE

“The authors wish to acknowledge the financial support that facilitated the research, writing, and publication of this paper. Specifically, YK received funding from the Scientific Research Grant (KAKENHI 21K21154), whereas HN was supported through the Scientific Research Grant (KAKENHI 23K09604).”

4. As required by our policy on Data Availability, please ensure your manuscript or supplementary information includes the following:

Additional Editor Comments:

Dear Authors,

I would like to commend you on the overall quality of the manuscript, which is well-written and addresses a topic of relevance. Based on the feedback received, two reviewers have suggested only minor revisions, while one has recommended rejection.

I believe that the suggestions provided by the reviewers are constructive and can be addressed effectively with some adjustments. I encourage you to carefully review the feedback and make the necessary revisions, as I think these changes will further improve the manuscript and enhance its chances of acceptance upon resubmission.

I look forward to seeing the revised version.

Best regards,

Dr Josep Vidal-Alaball

Reviewers' comments:

Reviewer's Responses to Questions

**Comments to the Author**

1. Is the manuscript technically sound, and do the data support the conclusions?

Reviewer #1: Yes

Reviewer #2: Partly

Reviewer #3: Yes

2. Has the statistical analysis been performed appropriately and rigorously? 

Reviewer #1: N/A

Reviewer #2: N/A

Reviewer #3: Yes

3. Have the authors made all data underlying the findings in their manuscript fully available?

Reviewer #1: Yes

Reviewer #2: Yes

Reviewer #3: Yes

4. Is the manuscript presented in an intelligible fashion and written in standard English?

Reviewer #1: Yes

Reviewer #2: Yes

Reviewer #3: Yes

5. Review Comments to the Author

Reviewer #1: I really enjoyed reading this manuscript. The idea of scoping review on such a less researched area was interesting and can provide insightful findings.

There are some points that can help improve the paper:

1. in review papers, it is common to mention the reason(s) why the criteria regarding the study quality for selection of papers are the way they are. it is recommended to mention on what basis such criteria are adopted as the generalizations of findings are under question.

2. it is suggested to include more articles as the number of articles (i.e., 15 articles) may not provide a comprehensive picture of what is going on in that area. So, if it is possible, widen the scope of this scoping review. As it was mentioned in the manuscript, out of 1,484, only 15 met our strict inclusion criteria.

3. because of the low number of the studies, in the discussion section, biases in the number of studies are explained on and because of the strict criteria, the discussion cannot be convincing.

the method section was rigorous as the the degree of agreement was mentioned and coding seems to have been done correctly. In the method, only the inclusion criteria is recommended to change, if possible.

the APA was observed. Furthermore, the manuscript was smooth to follow and erroneous grammatical structures were not observed.

Reviewer #2: Although the topic [Current status of tele-speech language therapy by type and support for patients with

post-stroke aphasia: a scoping review] is quite interesting, the article is only a review paper offering very limited insights which I believe are not significant enough to be considered for publication at a journal like PlosOne. The authors could perhaps consider either adding experimental data / case studies, or submitting the paper to a lower level journal.

Reviewer #3: Limited to English and Japanese language studies, potentially missing relevant research in other languages

The methods section is generally well-described. The use of the PRISMA-ScR guidelines is appropriate. However, the authors could provide more detail on how disagreements in the screening process were resolved.

Consider including a flow diagram of the study selection process (PRISMA flow diagram)

Provide more details on the data extraction process and how data were synthesized

Discuss the potential impact of excluding non-English and non-Japanese studies on the review's findings

6. PLOS authors have the option to publish the peer review history of their article (what does this mean? ). If published, this will include your full peer review and any attached files.

**Do you want your identity to be public for this peer review?** For information about this choice, including consent withdrawal, please see our Privacy Policy .

Reviewer #1: No

Reviewer #2: No

Reviewer #3: **Yes: ** Yasser Alrefaee

---

## [Author Response · Author response to Decision Letter 0]

22 Nov 2024

Reviewer 1

1. in review papers, it is common to mention the reason(s) why the criteria regarding the study quality for selection of papers are the way they are. It is recommended to mention on what basis such criteria are adopted as the generalizations of findings are under question.

Response to the reviewer: Thank you for your valuable feedback regarding our research quality selection criteria. We recognize that the rationale for the selection criteria described in the Methods section requires further clarification. To address this, we have revised the "Selection Criteria" subsection in the Methods section (Pages 6–7: Lines 105–112).

“Selection criteria

We included studies that investigated the assessment or training methods of Tele-SLT for PWA following a stroke. Studies were excluded if they: 1) involved participants under 18 years of age; 2) combined Tele-SLT with in-person SLT; or 3) were review articles (e.g., systematic reviews, scoping reviews, narrative reviews), case reports, qualitative studies, cost-effectiveness analyses, books, conference proceedings, or study protocols. Full details of the inclusion and exclusion criteria are provided in Table 1.”

2. it is suggested to include more articles as the number of articles (i.e., 15 articles) may not provide a comprehensive picture of what is going on in that area. So, if it is possible, widen the scope of this scoping review. As it was mentioned in the manuscript, out of 1,484, only 15 met our strict inclusion criteria.

Response to the reviewer: Thank you for your insightful comments regarding the comprehensiveness of the review. In response, we have revised the study selection criteria to encompass a broader range of research trends. The following changes were made:

1. We removed the restriction on the minimum sample size (n = 20) and reevaluated the selection criteria. This adjustment allowed the inclusion of smaller-scale studies that were previously excluded. Consequently, 20 additional studies were incorporated, increasing the total from 15 to 35—a 133% increase.

2. Despite expanding the scope of the analysis, as detailed in the Methods (Pages 6–10, Lines 89–163) and Results (Pages 10–31, Lines 165–281), the key research trends in tele-rehabilitation for speech-language therapy remain consistent. This consistency further reinforces the reliability of our findings.

3. because of the low number of the studies, in the discussion section, biases in the number of studies are explained on and because of the strict criteria, the discussion cannot be convincing.

the method section was rigorous as the the degree of agreement was mentioned and coding seems to have been done correctly. In the method, only the inclusion criteria is recommended to change, if possible.

Response to the reviewer: Thank you for your thoughtful evaluation of the rigor of our methodology. We agree with your observation that while the reliability of the methodology was ensured through stringent measures, the strict inclusion criteria may have narrowed the scope of the analysis and reduced the persuasiveness of the discussion. To address this issue, we have made the following improvements:

1. As detailed in the Methods section (Pages 6–10, Lines 89–163), we removed the sample size criterion (n ≥ 20) and included smaller-scale studies in our analysis. This revision increased the number of studies from 15 to 35 (Fig 1, Page 11).

2. In the Results section (Pages 10–31, Lines 165–280), we have added a detailed description of the findings based on the expanded dataset.

3. In the Discussion section (Pages 32–38, Lines 291–389), we broadened our interpretation to reflect the expanded evidence base and provided a more thorough examination of the generalizability of the findings.

Additionally, we have preserved the rigor of other methodological aspects, such as the reliability of the coding process, in accordance with established procedures. 

Reviewer 2

Although the topic [Current status of tele-speech language therapy by type and support for patients with post-stroke aphasia: a scoping review] is quite interesting, the article is only a review paper offering very limited insights which I believe are not significant enough to be considered for publication at a journal like PlosOne. The authors could perhaps consider either adding experimental data / case studies, or submitting the paper to a lower level journal.

Response to the reviewer: Thank you for your thoughtful feedback. We have addressed your concerns by expanding our study scope and conducting a comprehensive additional literature review. As a result, 20 new studies were included, increasing the total to 35. We also performed an exploratory meta-analysis on the training effects of telepractice speech-language therapy (Tele-SLT). The 20 additional papers are listed in Tables and Fig. 1, while the results of the meta-analysis are detailed in the "Training Effects of Tele-SLT" section of the Results (Page 31, Lines 265–280, Figs. 4–6).

Our findings revealed that asynchronous training positively impacted language functions in individuals with aphasia, whereas synchronous training did not show a significant effect on communication skills. Based on these insights, we have significantly enhanced the discussion to provide a more comprehensive analysis.

A scoping review including 35 studies on support for individuals with aphasia is uncommon, even among recent similar studies (e.g., Currie, S. S., et al., 2024; Hernandez, N. J., et al., 2024). Furthermore, meta-analyses specifically focused on Tele-SLT training remain extremely limited. We believe this study offers a valuable contribution by providing a broader and more detailed analysis compared to prior research. We sincerely hope that the revisions meet your expectations.

“Training effects of Tele-SLT (Page 31, Lines 265–280)

We performed a meta-analysis of Tele-SLT training effects using data from four RCTs. Two studies examined synchronous training using the CAL and CETI as outcome measures [38, 41], and two examined asynchronous training using the WAB as the outcome measure [44, 55]. For synchronous training, there was no significant difference between the intervention and control groups in combined CAL and CETI scores (fixed-effects model, SMD = -0.10, 95% CI = -0.57 to 0.37, P > 0.05) (Figure 4). For asynchronous training, analysis of the WAB-AQ showed no significant difference between groups (fixed-effects model, MD = 8.55, 95% CI = -4.59 to 21.70, P > 0.05) (Figure 5). However, a significant difference was found in the combined WAB subscales (Comprehension, Naming, Repetition) (fixed-effects model, MD = 0.70, 95% CI = 0.03 to 1.37, P < 0.05). Despite the significant result for the combined subscales, no significant differences were observed for individual subscales: Comprehension (fixed-effects model, MD = 0.46, 95% CI = -0.42 to 1.34, P > 0.05), Naming (fixed-effects model, MD = 1.21, 95% CI = -0.23 to 2.65, P > 0.05), and Repetition (fixed-effects model, MD = 0.83, 95% CI = -0.61 to 2.28, P > 0.05) (Fig 6)” 

Reviewer 3

1. Limited to English and Japanese language studies, potentially missing relevant research in other languages

Response to the reviewer: Thank you for raising these important points.

In this study, we followed the methodology of prior reviews in the field of aphasia (e.g., Wang, G., et al., 2020) and limited our scope to studies written in the authors' native language, Japanese, and in English, which encompass a significant portion of relevant research. However, as you rightly pointed out, excluding studies published in other languages may have introduced selection bias.

Although directly addressing this issue within the scope of this study is challenging, we have acknowledged it as a limitation in the Limitations section (Page 37, Lines 383–385).

“Second, while we performed a comprehensive literature search, the exclusion of non-English and non-Japanese articles and the absence of hand-searching of grey literature or specific databases may have introduced selection bias.”

2. The methods section is generally well-described. The use of the PRISMA-ScR guidelines is appropriate. However, the authors could provide more detail on how disagreements in the screening process were resolved.

Response to the reviewer: Thank you very much for your valuable feedback. We appreciate your recognition of the appropriateness of the Methods section and the application of the PRISMA-ScR guidelines.

As you correctly pointed out, our original description lacked sufficient detail on how disagreements during the screening process were resolved. To address this, we have revised the "Screening Method" and "Data Extraction Process" subsections in the Methods section to provide a more detailed explanation (Pages 8–9, Lines 115–136).

“Screening method

Two authors (YK and SN) independently screened titles and abstracts of identified articles. Subsequently, two other authors (YK and AY) independently reviewed the full texts of the selected articles for inclusion. Any discrepancies between authors during the screening process were resolved through discussion. If a consensus could not be reached, a third author (HN) made the final decision.

Data extraction process

Two authors (YK and AY) independently reviewed the full texts of articles selected in the second screening and extracted data. Discrepancies in data extraction were resolved through discussion between the two authors. If consensus could not be reached, a third author (HN) made the final decision. For studies on tele-assessment methods, the following information was collected: author, year of publication, country of the first author, participant characteristics (e.g., gender, age), number of participants, domains covered (e.g., language function, communication, well-being, QOL), software used, electronic devices used, and scale accuracy (e.g., reliability, validity). For studies on tele-training methods, the following information was collected: author, year of publication, study design, participant characteristics (gender, age), number of participants, domains covered (language function, communication, well-being, QOL), software used, electronic devices used, and training outcomes. One author (YK) managed the data using Rayyan reference management software and compiled the data into tables using Excel. Any missing articles were obtained by contacting the corresponding authors.”

3. Consider including a flow diagram of the study selection process (PRISMA flow diagram)

Response to the reviewer: Thank you very much for your valuable suggestion.

We agree that including a PRISMA flow diagram is essential for clearly presenting the study selection process. In response, we have incorporated a PRISMA flow diagram into the manuscript (Fig. 1, Page 11), which visually depicts the literature search and selection process. We believe this addition significantly enhances the clarity, transparency, and reproducibility of our study.

Once again, we sincerely appreciate your insightful advice.

4. Provide more details on the data extraction process and how data were synthesized

Response to the reviewer: We sincerely thank you for your valuable feedback.

We agree that our explanation of the data extraction process and data integration methods was insufficient for readers. In response to your suggestion, we have added further details to the "Data Extraction Process" subsection in the Methods section and revised it accordingly (Pages 8–9, Lines 121–136). Additionally, for the 90 papers subjected to secondary screening, both the evaluator's ratings and the final ratings are now presented in Table S3.

Once again, we greatly appreciate your insightful suggestions, which have helped us improve the clarity and transparency of our study.

“Data extraction process

Two authors (YK and AY) independently reviewed the full texts of articles selected in the second screening and extracted data. Discrepancies in data extraction were resolved through discussion between the two authors. If consensus could not be reached, a third author (HN) made the final decision. For studies on tele-assessment methods, the following information was collected: author, year of publication, country of the first author, participant characteristics (e.g., gender, age), number of participants, domains covered (e.g., language function, communication, well-being, QOL), software used, electronic devices used, and scale accuracy (e.g., reliability, validity). For studies on tele-training methods, the following information was collected: author, year of publication, study design, participant characteristics (gender, age), number of participants, domains covered (language function, communication, well-being, QOL), software used, electronic devices used, and training outcomes. One author (YK) managed the data using Rayyan reference management software and compiled the data into tables using Excel. Any missing articles were obtained by contacting the corresponding authors.

5. Discuss the potential impact of excluding non-English and non-Japanese studies on the review's findings

Response to the reviewer: Thank you very much for your valuable feedback.

As you rightly pointed out, our discussion did not sufficiently address the potential impact of excluding studies published in languages other than English and Japanese on the results of this review. To address this, we have added a detailed discussion of this issue in the Limitations section (Pages 37–38, Lines 380–389).

We sincerely appreciate your insight, which has helped us enhance the depth and transparency of our discussion.

“Second, while we performed a comprehensive literature search, the exclusion of non-English and non-Japanese articles and the absence of hand-searching of grey literature or specific databases may have introduced selection bias.”

---

## [Decision Letter · Decision Letter 1]

5 Jan 2025

PONE-D-24-13841R1

Current status of tele-speech language therapy by type and support for patients with post-stroke aphasia: a scoping review

PLOS ONE

Dear Dr. Nagami,

Thank you for submitting your manuscript to PLOS ONE. After careful consideration, we feel that it has merit but does not fully meet PLOS ONE’s publication criteria as it currently stands. Therefore, we invite you to submit a revised version of the manuscript that addresses the points raised during the review process.

We look forward to receiving your revised manuscript.

Kind regards,

Josep Vidal-Alaball, MD, PdD, MPH

Academic Editor

PLOS ONE

Journal Requirements:

Reviewers' comments:

Reviewer's Responses to Questions

**Comments to the Author**

1. If the authors have adequately addressed your comments raised in a previous round of review and you feel that this manuscript is now acceptable for publication, you may indicate that here to bypass the “Comments to the Author” section, enter your conflict of interest statement in the “Confidential to Editor” section, and submit your "Accept" recommendation.

Reviewer #1: All comments have been addressed

2. Is the manuscript technically sound, and do the data support the conclusions?

Reviewer #1: Yes

3. Has the statistical analysis been performed appropriately and rigorously? 

Reviewer #1: Yes

4. Have the authors made all data underlying the findings in their manuscript fully available?

Reviewer #1: Yes

5. Is the manuscript presented in an intelligible fashion and written in standard English?

Reviewer #1: Yes

6. Review Comments to the Author

Reviewer #1: The manuscript has significantly improved, and the authors have addressed all the comments provided. The number of articles has increased, which offers more insightful information. Additionally, a meta-analysis has been incorporated into the study. However, the scope remains limited, as the focus was primarily on two languages and the inclusion criteria were quite strict. There is also potential for an increase in the number of studies included in the meta-analysis.

7. PLOS authors have the option to publish the peer review history of their article (what does this mean? ). If published, this will include your full peer review and any attached files.

**Do you want your identity to be public for this peer review?** For information about this choice, including consent withdrawal, please see our Privacy Policy .

Reviewer #1: No

---

## [Author Response · Author response to Decision Letter 1]

31 Jan 2025

Response to the reviewer

Comment 1:

The manuscript has significantly improved, and the authors have addressed all the comments provided. The number of articles has increased, which offers more insightful information. Additionally, a meta-analysis has been incorporated into the study. However, the scope remains limited, as the focus was primarily on two languages and the inclusion criteria were quite strict. There is also potential for an increase in the number of studies included in the meta-analysis.

Response to the reviewer: Thank you for your valuable feedback on the revised manuscript, which includes additional studies and new analyses. We appreciate your insightful comments regarding the meta-analysis. Our initial stringent inclusion criteria resulted in a small sample size. Therefore, following your suggestion, we incorporated the Western Aphasia Battery-Revised (WAB-R) as an outcome measure and included the results from Braley M., et al. [62] in the analysis of asynchronous speech-language therapy (SLT).

While the addition of the WAB-R and the study by Braley et al. did not reveal a significant effect of asynchronous SLT on the Aphasia Quotient or the subtests of either the Western Aphasia Battery or the WAB-R, we did observe an overall effect. This finding remains consistent with our previous conclusion and is reflected in the Results section and Figures 5 and 6.

We acknowledge the limitations of the meta-analysis due to the small number of included studies and the potential lack of optimal standardization regarding inclusion criteria, outcome measures, and interventions provided to the control groups. We have added a statement addressing these limitations to the "Study Limitations" subsection of the Discussion.

(Page 31–32, Line 267–284)

Training effects of Tele-SLT

We performed a meta-analysis of Tele-SLT training effects using data from four RCTs. Two studies examined synchronous training using the CAL and CETI as outcome measures [39, 42], and three examined asynchronous training using the WAB, WAB-R as the outcome measure [45, 56, 62]. For synchronous training, there was no significant difference between the intervention and control groups in combined CAL and CETI scores (fixed-effects model, SMD = -0.10, 95% CI = -0.57 to 0.37, P > 0.05) (Figure 4). For asynchronous training, analysis of the WAB-AQ showed no significant difference between groups (fixed-effects model, SMD = 0.24, 95% CI = -0.17 to 0.65, P > 0.05) (Figure 5). However, a significant difference was found in the combined WAB subscales (Spontaneous speech, Comprehension, Repetition, Naming) (fixed-effects model, SMD = 0.23, 95% CI = 0.02 to 0.44, P < 0.05). Despite the significant result for the combined subscales, no significant differences were observed for individual subscales: Spontaneous speech (fixed-effects model, SMD = 0.20, 95% CI = -0.26 to 0.67, P > 0.05), Comprehension (fixed-effects model, SMD = 0.28, 95% CI = -0.13 to 0.69, P > 0.05), Repetition (fixed-effects model, SMD = 0.13, 95% CI = -0.28 to 0.54, P > 0.05) and Naming (fixed-effects model, SMD = 0.31, 95% CI = -0.11 to 0.72, P > 0.05) (Fig 6)

(Page 37–38, Line 385–398)

Limitations of this Review

This review has five main limitations. First, the failure to use specific checklists for the quantitative quality assessment of the included studies may introduce uncertainty regarding the consistency of the study quality in this scoping review design. Second, while a comprehensive literature search was conducted, the exclusion of articles in languages other than English and Japanese, as well as gray literature and manual searches of specific databases, may have led to selection bias. Third, the exclusion of articles in languages other than English and Japanese in the meta-analysis limited the number of studies included for analysis, and we could not conduct subgroup analyses based on participant characteristics or details of interventions. Therefore, caution is needed when interpreting the results of the meta-analysis. Fourth, as we focused on studies that included multiple participants with aphasia, single-case study designs were excluded. Finally, as with all database searches, it is impossible to completely eliminate publication bias. [78]

---

## [Editor Report · Decision Letter 2]

4 Feb 2025

PONE-D-24-13841R2Current status of tele-speech language therapy by type and support for patients with post-stroke aphasia: a scoping reviewPLOS ONE

Dear Dr. Nagami,

Thank you for submitting your manuscript to PLOS ONE. After careful consideration, we feel that it has merit but does not fully meet PLOS ONE’s publication criteria as it currently stands. Therefore, we invite you to submit a revised version of the manuscript that addresses the points raised during the review process.

We look forward to receiving your revised manuscript.

Kind regards,

Josep Vidal-Alaball, MD, PdD, MPH

Academic Editor

PLOS ONE

Journal Requirements:

Additional Editor Comments :

Please address the reviewers' comments (reviewer 3).
---

## [Author Response · Author response to Decision Letter 2]

6 Feb 2025

Response to the reviewer

Comment 1:

The manuscript has significantly improved, and the authors have addressed all the comments provided. The number of articles has increased, which offers more insightful information. Additionally, a meta-analysis has been incorporated into the study. However, the scope remains limited, as the focus was primarily on two languages and the inclusion criteria were quite strict. There is also potential for an increase in the number of studies included in the meta-analysis.

Response to the reviewer: Thank you for your valuable feedback on the revised manuscript, which includes additional studies and new analyses. We appreciate your insightful comments regarding the meta-analysis. Our initial stringent inclusion criteria resulted in a small sample size. Therefore, following your suggestion, we incorporated the Western Aphasia Battery-Revised (WAB-R) as an outcome measure and included the results from Braley M., et al. [62] in the analysis of asynchronous speech-language therapy (SLT).

While the addition of the WAB-R and the study by Braley et al. did not reveal a significant effect of asynchronous SLT on the Aphasia Quotient or the subtests of either the Western Aphasia Battery or the WAB-R, we did observe an overall effect. This finding remains consistent with our previous conclusion and is reflected in the Results section and Figures 5 and 6.

We acknowledge the limitations of the meta-analysis due to the small number of included studies and the potential lack of optimal standardization regarding inclusion criteria, outcome measures, and interventions provided to the control groups. We have added a statement addressing these limitations to the "Study Limitations" subsection of the Discussion.

(Page 31–32, Line 267–284)

Training effects of Tele-SLT

We performed a meta-analysis of Tele-SLT training effects using data from four RCTs. Two studies examined synchronous training using the CAL and CETI as outcome measures [39, 42], and three examined asynchronous training using the WAB, WAB-R as the outcome measure [45, 56, 62]. For synchronous training, there was no significant difference between the intervention and control groups in combined CAL and CETI scores (fixed-effects model, SMD = -0.10, 95% CI = -0.57 to 0.37, P > 0.05) (Figure 4). For asynchronous training, analysis of the WAB-AQ showed no significant difference between groups (fixed-effects model, SMD = 0.24, 95% CI = -0.17 to 0.65, P > 0.05) (Figure 5). However, a significant difference was found in the combined WAB subscales (Spontaneous speech, Comprehension, Repetition, Naming) (fixed-effects model, SMD = 0.23, 95% CI = 0.02 to 0.44, P < 0.05). Despite the significant result for the combined subscales, no significant differences were observed for individual subscales: Spontaneous speech (fixed-effects model, SMD = 0.20, 95% CI = -0.26 to 0.67, P > 0.05), Comprehension (fixed-effects model, SMD = 0.28, 95% CI = -0.13 to 0.69, P > 0.05), Repetition (fixed-effects model, SMD = 0.13, 95% CI = -0.28 to 0.54, P > 0.05) and Naming (fixed-effects model, SMD = 0.31, 95% CI = -0.11 to 0.72, P > 0.05) (Fig 6)

(Page 37–38, Line 385–398)

Limitations of this Review

This review has five main limitations. First, the failure to use specific checklists for the quantitative quality assessment of the included studies may introduce uncertainty regarding the consistency of the study quality in this scoping review design. Second, while a comprehensive literature search was conducted, the exclusion of articles in languages other than English and Japanese, as well as gray literature and manual searches of specific databases, may have led to selection bias. Third, the exclusion of articles in languages other than English and Japanese in the meta-analysis limited the number of studies included for analysis, and we could not conduct subgroup analyses based on participant characteristics or details of interventions. Therefore, caution is needed when interpreting the results of the meta-analysis. Fourth, as we focused on studies that included multiple participants with aphasia, single-case study designs were excluded. Finally, as with all database searches, it is impossible to completely eliminate publication bias. [78]

---

## [Editor Report · Decision Letter 3]

9 Feb 2025

Current status of tele-speech language therapy by type and support for patients with post-stroke aphasia: a scoping review

PONE-D-24-13841R3

Dear Dr. Nagami,

We’re pleased to inform you that your manuscript has been judged scientifically suitable for publication and will be formally accepted for publication once it meets all outstanding technical requirements.

Kind regards,

Josep Vidal-Alaball, MD, PdD, MPH

Academic Editor

PLOS ONE
---

## [Editor Report · Acceptance letter]

PONE-D-24-13841R3

PLOS ONE

Dear Dr. Nagami,

I'm pleased to inform you that your manuscript has been deemed suitable for publication in PLOS ONE. Congratulations! Your manuscript is now being handed over to our production team.

Kind regards,

on behalf of

Dr. Josep Vidal-Alaball

Academic Editor

PLOS ONE